# Chemistry and Pharmacological Activity of Sesquiterpenoids from the *Chrysanthemum* Genus

**DOI:** 10.3390/molecules26103038

**Published:** 2021-05-19

**Authors:** Sai Jiang, Mengyun Wang, Zichen Jiang, Salman Zafar, Qian Xie, Yupei Yang, Yang Liu, Hanwen Yuan, Yuqing Jian, Wei Wang

**Affiliations:** 1TCM and Ethnomedicine Innovation & Development International Laboratory, Innovative Materia Medica Research Institute, School of Pharmacy, Hunan University of Chinese Medicine, Changsha 410208, China; saijiang626@hotmail.com (S.J.); 20183362@stu.hnucm.edu.cn (M.W.); xieqian@stu.hnucm.edu.cn (Q.X.); yangyupei24@163.com (Y.Y.); liuyang611@hotmail.com (Y.L.); hanwyuan@hnucm.edu.cn (H.Y.); cpujyq2010@163.com (Y.J.); 2Division of Biological Sciences, University of California San Diego, San Diego, CA 95101, USA; z8jiang@ucsd.edu; 3Institute of Chemical Sciences, University of Peshawar, Peshawar 25120, Pakistan; salmanzafar@uop.edu.pk

**Keywords:** *Chrysanthemum* genus, sesquiterpenoids, biosynthetic pathway, pharmacological effects

## Abstract

Plants from the *Chrysanthemum* genus are rich sources of chemical diversity and, in recent years, have been the focus of research on natural products chemistry. Sesquiterpenoids are one of the major classes of chemical constituents reported from this genus. To date, more than 135 sesquiterpenoids have been isolated and identified from the whole genus. These include 26 germacrane-type, 26 eudesmane-type, 64 guaianolide-type, 4 bisabolane-type, and 15 other-type sesquiterpenoids. Pharmacological studies have proven the biological potential of sesquiterpenoids isolated from *Chrysanthemum* species, reporting anti-inflammatory, antibacterial, antitumor, insecticidal, and antiviral activities for these interesting molecules. In this paper, we provide information on the chemistry and bioactivity of sesquiterpenoids obtained from the *Chrysanthemum* genus which could be used as the scientific basis for their future development and utilization.

## 1. Introduction

The *Chrysanthemum* genus belongs to the family Compostae and is comprised of more than 40 accepted species of flowering plants (http://www.theplantlist.org/ accessed on 22 March 2021), which are perennial herbs mainly distributed in East Asia; more than 20 species are found in China [1,2]. The flowers of several *Chrysanthemum* species, including ornamental cultivars and hybrids are economically the second most important floricultural crop after rose, and one of the most important sources of flower arrangement and potted plants around the world [3,4]. In addition, the flower is regularly consumed in many countries as both food and medicine. Traditionally, in China and Japan, some *Chrysanthemum* species, such as *Chrysanthemum morifolium* Ramat. (Juhua in Chinese) and *Chrysanthemum indicum* L. (Yejuhua in Chinese), are used as sedative, anti-inflammatory, antitussive, and a general tonic. *C. morifolium* and *C. indicum* are slightly cold in effect with a sweet and bitter taste. *Chinese Pharmacopoeia* (2015 edition) records their good effects on liver and heart. They are effective in reducing heat and detoxification, treating swollen furuncle carbuncle, swelling and eye pain, headache, and dizziness [5,6]. Modern pharmacological studies have shown that the *Chrysanthemum* genus has antitumor, antioxidant, anti-inflammatory and antibacterial properties. The species have been found effective against cardiovascular diseases and for reducing fat and cholesterol contents in the blood [7,8,9,10]. With the traditional Chinese Medicine Fufang, *C. morifolium* granule, *C. indicum* granule, and *C. indicum* injections are used to treat prostatitis, chronic pelvic inflammation, and upper respiratory tract infection [6]. In addition to sesquiterpenoids, the *Chrysanthemum* genus is also known to be a rich source of terpenoids, phenylpropanoids, and flavonoids [11,12,13].

Sesquiterpenoids are mainly found in essential oils, detected by GC-MS analysis [14,15]. In fact, most of these compounds have been purified from aqueous ethanolic extract of *Chrysanthemum* plants through chromatography on different adsorbents such as silica gel, C_18_ silica (reversed-phase column), Sephadex LH-20, and through semi-preparative HPLC. All the sesquiterpenoids have been isolated from air-dried flowers or aerial parts of the *Chrysanthemum* genus, except angeloylcumambrin B (**53**), cumambrin A (**54**), cumambrin-B (**55**), and handelin (**104**) which were obtained from fresh whole herbs (Table 1). *C. morifolium*, *C. indicum*, *C lavandulifolium*, *C. zawadskii*, and *C. ornatum* have been reported to be rich in sesquiterpenoids, especially the *C. morifolium* and *C. indicum* species [16,17,18,19]. To the best of our knowledge, there is not a single comprehensive review on the sesquiterpenoids of the *Chrysanthemum* genus. This review article is an attempt to summarize the sesquiterpenoids of the *Chrysanthemum* genus, highlight their possible biosynthetic pathway and pharmacological effects, and provide a rationale for future development and research on this important genus.

## 2. Sesquiterpenoids

Sesquiterpenoids are compounds made up of three isoprene units, containing 15 carbons [20]. Some sesquiterpenoids with different skeleton types have been isolated and identified from the *Chrysanthemum* genus, including germacrane-type, eudesmane-type and guaianolide-type sesquiterpenoids. Interestingly, most of these compounds have been reported from just one species, except angeloylcumambrin B (**53**), cumambrin A (**54**), and handelin (**104**) which were reported from more than one species. The following is a classification of the sesquiterpenoids found in the *Chrysanthemum* genus. A complete profile of these compounds is given in Table 1.

### 2.1. Germacrane-Type Sesquiterpenoids

Germacrane-type sesquiterpenoids are monocyclic sesquiterpenoids, which are composed of a 10-membered carbon ring, a methyl group at C-4 and C-10, and an isopropyl group at C-7. Twenty-six compounds of this type have so far been reported from the genus. Compounds **1**–**13**, **15**, and **16** vary in the position and orientation of the oxygen-containing substituents. Compounds **14** and **17**–**24** form five-membered *γ*-lactone ring at C-6 and C-12. Compounds **25** and **26** form irregular rings due to the configuration of double bond in the ring. Examples of structures are shown in Figure 1.

### 2.2. Eudesmane-Type Sesquiterpenoids

Eudesmane-type sesquiterpenoids are bicyclic compounds, composed of two six-membered carbon rings with methyl groups at C-4 and C-10 positions, and an isopropyl group at C-7. Twenty-six compounds with this skeleton have been isolated from this genus, however, no eudesmanolides have been reported so far. Compounds **35** and **38** are rare due to α-CH_3_ at C-10, others having a β-CH_3_ at the same position. Examples of structures are shown in Figure 2.

### 2.3. Guaianolide-Type Sesquiterpenoids

Guaianolide-type sesquiterpenoids have ternary rings consisting of a five-membered ring, a seven-membered ring, a five-membered γ-lactone ring, and methyl groups at C-4 and C-10 positions. The structures are quite complicated with up to nine stereocenters, mostly highly oxidized. Moreover, they are also the most abundant compounds in the *Chrysanthemum* genus. Compound **90** contains chlorine atoms. So far, 60 chlorine-containing guaianolide-type sesquiterpenoids have been isolated from Compositae [39,46,47]. Examples of structures are given in Figure 3.

### 2.4. 1,10-Seco Guaianolide Sesquiterpenoids

1,10-Seco guaianolide sesquiterpenoids have a binary ring system. The seven-membered ring in the middle is disconnected at C-1 and C-10 of guaianolide-type sesquiterpenoids. Six compounds of this subclass have been reported from the genus under focus (Figure 4).

### 2.5. Disesquiterpenoids and a Trisesquiterpenoid

Disesquiterpenoids and a trisesquiterpenoid are composed of an electron donor conjugated diene and an electrophilic double bond fragment formed through [4 + 2] Diels–Alder reaction. An electron donor diene is a C5/C7/C5 guaianolide-type structure, while the electron deficient double bond is provided with the associated α-methylene-γ-lactone fragment [48]. Guaianolide-type compounds in the *Chrysanthemum* genus form dimers or trimers by cyclization to form five-membered ring. Examples of structures are given in Figure 5.

### 2.6. Other Types of Sesquiterpenoids

Nineteen other types of sesquiterpenoids have also been reported from this genus including, four bisabolane-type, one eremophilane-type, one cadinane-type, one oplopanane-type, one dodecane-type, and a carabranolide-type sesquiterpenoids (Figure 6).

## 3. Biosynthetic Pathway of Sesquiterpenoids

The main skeleton types of sesquiterpenoids in the *Chrysanthemum* genus are germacrane-type sesquiterpenoids, eudesmane-type sesquiterpenoids, and guaianolide-type sesquiterpenoids [49,50]. The proposed pathway to the sesquiterpenoids starts with the cyclization of farnesyl diphosphate (FPP) to germacrene A [50]. We consider (+)-germacrene A formation to be the first committed step in sesquiterpenoids biosynthesis. A direct cyclisation of FPP to a eudesmane or guaiane does not occur. We suggest that oxidations of the common (+)-germacrene A intermediate determine how additional cyclisation occurs, i.e., whether guaianolides or eudesmanes (plus germacranes) are biosynthesized (Figure 7) [51,52,53,54,55,56]. Furthermore, 1,10-seco guaianolides and disesquiterpenoids are special guaianolides which were isolated from the *Chrysanthemum* genus. A plausible biosynthetic pathway of 1,10-seco guaianolides and disesquiterpenoids is given in Figure 8 [57].

## 4. Pharmacological Activities

The main pharmacological activities of sesquiterpenoids are antitumor, anti-inflammatory, antibacterial and antiviral. Some of the monomers are very good antitumor and anti-inflammatory agents [58].

### 4.1. Antitumor Activity

Current studies have shown that sesquiterpenoids have low cytotoxic effects on normal cells while playing an anticancer role, and therefore more and more experts and scholars are focusing on the antitumor properties of sesquiterpenoids [59]. Among them, sesquiterpenoid lactone have strong antitumor activity. It has been found that *α*-methylene-*γ*-butyrolactone ring is the effective group with antitumor activity [60].

Six compounds, angeloylcumambrin B (**53**), chrysanolide G (**56**), tigloylcumambrin B (**57**), chrysanolide E (**106**), 8′-tigloylchrysanolide D (**107**), and 8,8′-ditigloylchrysanolide D (**109**) were isolated and identified from *C. indicum* by the Xu Jun research group. These six compounds showed multiple cytotoxic activities against four human nasopharyngeal carcinoma (NPC) cell lines (CNE1, CNE2, HONE-1, and SUNE-1) and one human intestinal epithelial cell line (HT-29). The IC_50_ value of some compounds was lower than that of the positive control drug, among which compound **109** had the strongest activity against cancer. It was found that compound **109** concentration-dependently induces G2/M cell arrest. Thus, investigation of the mechanism of action and structure-activity relation (SAR) for compound **109** is worth exploring further [31]. In addition, the Kong Lingyi research group found that the new compound Chrysanthemulide A (CA) (**92**) demonstrated significant anti-osteosarcoma potential; therefore, its mechanism was studied. The c-Jun N-terminal kinase (JNK) signaling pathway was activated by CA, and treatment with JNK siRNAs or inhibitor SP600125 significantly reduced CA-mediated autophagosome accumulation and DR5-mediated cell apoptosis. CA can induce apoptosis according to upregulated DR5 via JNK-mediated autophagosome accumulation and the combined treatment of CA and TRAIL might be a promising therapy for osteosarcoma [61].

### 4.2. Anti-Inflammatory Activity 

The flower of *C. morifolium* and *C. indicum* is a common functional food and a well-known traditional Chinese medicine (TCM) for the treatment of inflammatory diseases [62]. The sesquiterpenoids isolated from the *Chrysanthemum* genus also showed strong anti-inflammatory activity. The anti-inflammatory effects on lipopolysaccharide (LPS)-induced nitric oxide (NO) were investigated in RAW 264.7 cells. Chrysanthemulide H (**86**), 8-tigloyldesacetylezomontanin (**87**), chrysanthemulide F (**89**), chrysanthemulide G (**90**), chrysanthemulide A (**92**), chrysanthemulide B (**93**), chrysanthemulide C (**94**), chrysanthemulide D (**95**), and chrysanthemulide E (**96**) and **98**–**103** displayed NO production inhibitory activities with IC_50_ values ranging from 1.4 to 9.7 μM. The IC_50_ value of the positive control drug L-NMMA was 25.8 μM [39,61]. These sesquiterpenoids have great anti-inflammatory potential. A mechanistic study revealed that the potential anti-inflammatory activity of compound chrysanthemulide A (**92**) appears to be mediated via suppression of an LPS-induced NF-κB pathway and downregulation of MAPK activation [39,61]. In addition, using H9c2 cardiocytes impaired by lipopolysaccharide (LPS), compounds chrysanthguaianolactone D (**65**), 3*α*,4*α*,10*β*-trihydroxy-8*α*-acetoxyguai-1,11(13)-dien-6*α*,12-olide (**67**), 3*α*,4*α*,10*β*-trihydroxy-8*α*-acetoxy-11*β*H-guai-1-en-6*α*,12-olide (**68**), 8*α*-(angelyloxy)-3*β*,4*β*-dihydroxy-5*α*H,6*β*H,7*α*H,11*α*H-guai-1(10)-en-12,6-olide (**71**), and 3*β*,4*α*-dihydroxy-8*α*-angelyloxy-1(10),11(13)-dien-6*β*,12-olide (**75**) exhibited anti-inflammatory activity [33].

### 4.3. Antibacterial Activity 

Two sesquiterpenoids were isolated and purified from the flower of *C zawadskii* and identified as angeloycumambrin B (**53**) and cumambrin A (**54**). Compared to the positive control benzoic acid and sorbic acid which showed antibacterial activity against *Bacillus subtilis*, *Staphylococcus aureus,* and *Vibrio parahemolyticus* with 10–13 mm diameter of clear zone (500 μg/disk), compound **53** (100 μg/disk) showed antibacterial activity against *B. subtilis*, *S. aureus,* and *V. parahemolyticus* with diameter of clear zone 12, 10, and 11 mm, respectively, and compound **54** (100 μg/disk) exhibited the activity against *B. subtilis* and *V. parahemolyticus* with 13 and 12 mm diameters of clear zone, respectively. Compounds **53** and **54** showed about five-fold stronger antibacterial activity against *B. subtilis* and *V. parahemolyticus* [30].

### 4.4. Antiviral Activity

There have been some compounds from *Chrysanthemum* species showing selective antiviral activities. Ten sesquiterpenoids, chrysanthemumin A (**31**), chrysanthemumin B (**35**), chrysanthemumin C (**42**), chrysanthemumin D (**33**), chrysanthemumin E (**34**), 6,8-cycloeudesm-4(15)-en-1-ol (**128**), 11-hydroxy-1-oxo-4*α*,5*α*,7*β*,10*β*-eremophilane (**121**), chrysanthediol A (**9**), 1*β*-hydroxy-4(15),5E,10(14)-germacratriene (**13**), and eudesm-4(15)-ene-1*β*,6*α*-diol (**52**) were isolated from *C. indicum* by the Shi Yanping research group. These compounds inhibited the porcine epidemic diarrhea virus (PEDV) protein expression, which showed that these compounds increased cell viability against cell death in PEDV-injected cells, among which compounds **13**, **35**, and **128** can also dose-dependently inhibite PEDV replication at concentrations ranging from 20 to 90 μM. The antiviral mechanism of compound **35** was studied. The results indicated that compound **35** exhibited potential inhibition of the viral protein synthesis in a dose-dependent manner [11]. In addition, chrysanolide B (**60**), chrysanolide C (**115**), and chrysanolide A (**116**) were evaluated for their anti-HBV activities, and 3TC (lamivudine, a frequently used clinical anti-HBV agent) was used as the positive control. These three compounds exhibited potent inhibitory activities against the secretion of HBsAg (IC_50_ = 131.28, 33.91, and 6.67 μM) and HBeAg (IC_50_ = 144.48, 30.09, and 6.23 μM), respectively. The positive control drug lamivudine had an IC_50_ value of 14.85 μM against HBsAg and 42.36 μM against HBeAg, respectively. More interestingly the anti-HBV activities increased with the increasing degree of aggregation. Compound **116** is a third guaianolide-type sesquiterpenoid trimer found in plant, its activity is better than that of the positive control drug, and its mechanism of anti-HBV is worthy of further exploration [43].

### 4.5. Antidiabetic and Antiobesity Activity 

10*α*-Hydroxy-1*α*,4*α*-endoperoxy-guaia-2-en-12,6*α*-olide (**88**) showed strong inhibitory effects against *α*-glucosidase and lipase activities, with IC_50_ values of 229.3 and 161.0 μM, respectively. The positive control drug acarbose inhibited α-glucosidase with IC_50_ value of 1907 μM, and orlistat inhibited lipase with IC_50_ value of 108.3 μM [40].

## 5. Conclusions

Sesquiterpenoids are one of the main chemical constituents of the *Chrysanthemum* genus which have various chiral centers and configurations. At present, 135 sesquiterpenes have been reported from the *Chrysanthemum* genus (most of them were isolated from *C. morifolium* and *C. indicum*), mainly including germacrane-type sesquiterpenoids, eudesmane-type sesquiterpenoids, and guaianolide-type sesquiterpenoids. The compounds have been found to exhibit wide spectrum pharmacological activities such as anti-inflammatory, antitumor, antibacterial, antiviral, antidiabetic, and anti-obesity activities. The activity of some compounds has been found to be better than that of positive control drugs in vitro. Some compounds, especially guaianolide-type sesquiterpenoids, showed better anticancer and anti-inflammatory activity, and their mechanism of action was explained from the aspects of signal pathway and target, which was consistent with the traditional function of clearing heat and removing toxicity exhibited by the *Chrysanthemum* genus, which is widely distributed in China and has unique advantages in number and species. Therefore, further research and development of the *Chrysanthemum* genus based on sesquiterpenoids need to be conducted.

At present, some progress has been made in the study of the sesquiterpenoids of the *Chrysanthemum* genus, but there are still some species to be explored. Most of the sesquiterpenoids have complicated structures, in which there are many chiral centers, conformers, and configurational isomers, and therefore it is difficult to separate and identify them. In terms of pharmacological activities, studies have mainly focused on the in vitro activity screening of pure compounds, while studies on pharmacodynamic evaluation and structure-activity relationship in vivo are almost nonexistent. However, it may be related to the insufficient yield and configuration of isolated compounds.

Therefore, for further research and development of the *Chrysanthemum* genus the following suggestions should be considered: Firstly, focus on other species of the *Chrysanthemum* genus that have not been fully studied, especially those whose chemical composition has not been studied. Secondly, try to obtain more sesquiterpenoids from *Chrysanthemum* genus plants by using more advanced methods. Thirdly, screen the sesquiterpenoids with strong activity and carry out synthesis and structural modification to study structure-activity relationships, efficacy evaluation, and mechanism targets in vivo, and to develop drugs for clinical use.

## Figures and Tables

**Figure 1 molecules-26-03038-f001:**
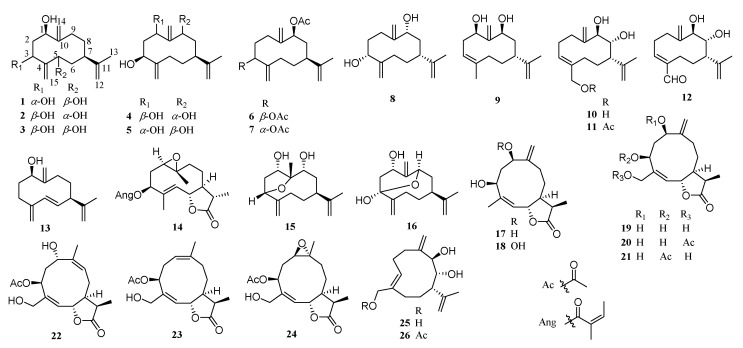
Germacrane-type sesquiterpenoids from *Chrysanthemum* genus.

**Figure 2 molecules-26-03038-f002:**
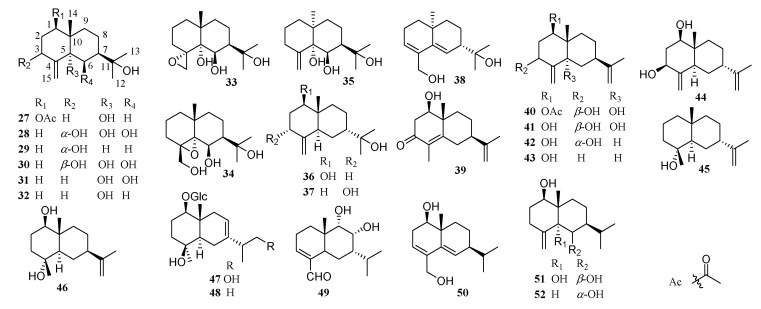
Eudesmane-type sesquiterpenoids from *Chrysanthemum* genus.

**Figure 3 molecules-26-03038-f003:**
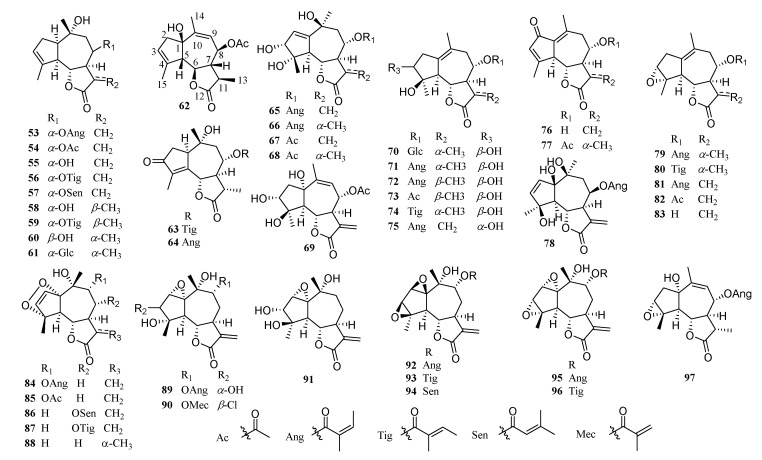
Guaianolide-type sesquiterpenoids from the *Chrysanthemum* genus.

**Figure 4 molecules-26-03038-f004:**
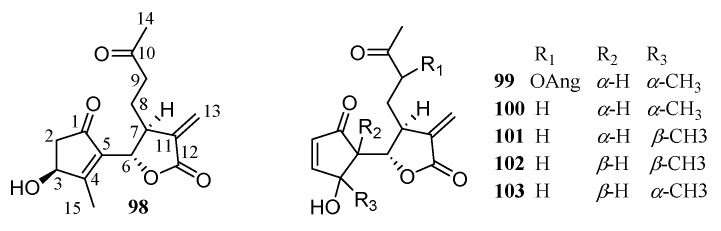
1,10-Seco guaianolides from the *Chrysanthemum* genus.

**Figure 5 molecules-26-03038-f005:**
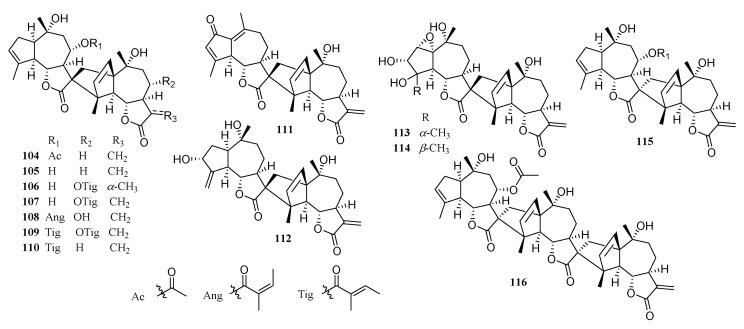
Disesquiterpenoids and a trisesquiterpenoid from the *Chrysanthemum* genus.

**Figure 6 molecules-26-03038-f006:**
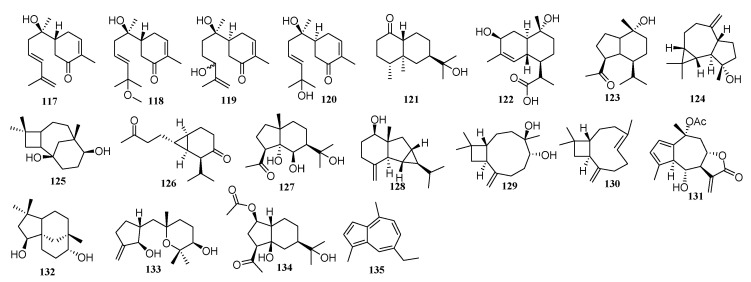
Other types of sesquiterpenoids from the *Chrysanthemum* genus.

**Figure 7 molecules-26-03038-f007:**
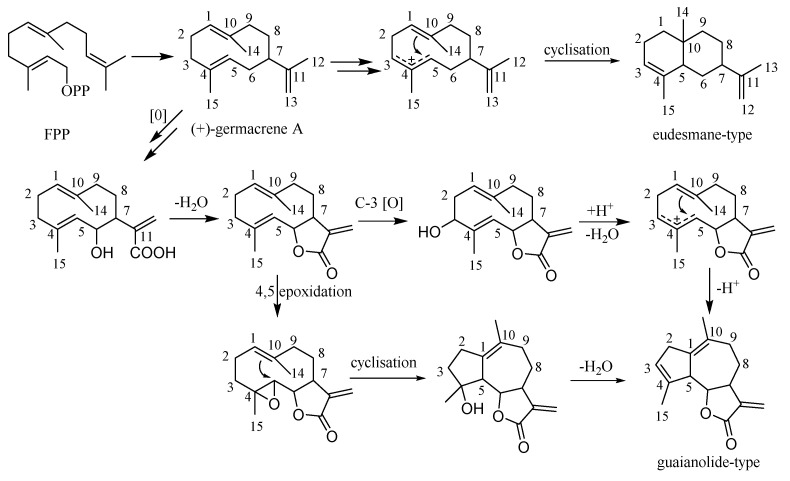
The connection of carbon skeleton of three sesquiterpenoids.

**Figure 8 molecules-26-03038-f008:**
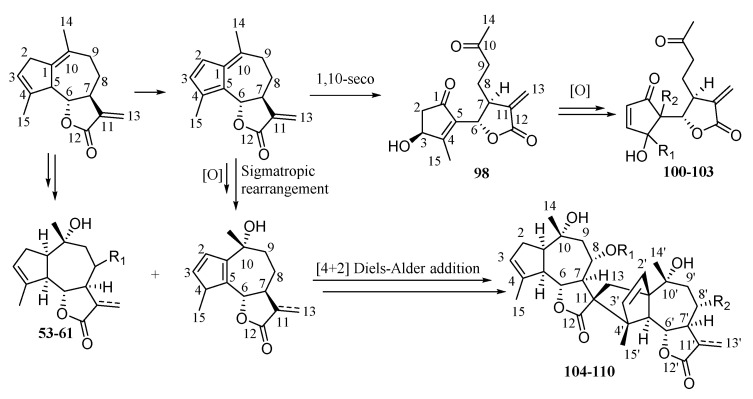
The plausible biosynthetic pathway of 1,10-seco guaianolides and disesquiterpenoids.

**Table 1 molecules-26-03038-t001:** Sesquiterpenoids from the *Chrysanthemum* genus.

No.	Compounds	Species	Parts Used	Identification Methods	Ref.
**Germacrane**					
**1**	1*β*,3*α*,5*β*-trihydroxyl-7-isopropenyl-germacren-4(15),10(14)-diene	*C. indicum*	Flowers	1D, 2D NMR; HRESIMS; X-ray	[16]
**2**	1*β*,3*β*,5*α*-trihydroxyl-7-isopropenyl-germacren-4(15),10(14)-diene	*C. indicum*	Flowers	1D, 2D NMR; HRESIMS	[16]
**3**	1*β*,3*β*,5*β*-trihydroxyl-7-isopropenyl-germacren-4(15),10(14)-diene	*C. indicum*	Flowers	1D, 2D NMR; HRESIMS	[16]
**4**	chrysanthemumin C	*C. indicum*	Flowers	1D, 2D NMR; HRESIMS	[21]
**5**	chrysanthemumin D	*C. indicum*	Flowers	1D, 2D NMR; HRESIMS	[21]
**6**	chrysanthediacetate B	*C. morifolium*	Flowers	1D, 2D NMR; EIMS	[17]
**7**	chrysanthediacetate C	*C. morifolium*	Flowers	1D, 2D NMR; EIMS	[17]
**8**	(3*R*,7*R*,9*R*)-3,9-dihydroxygermacra-4(15),10(14),11(12)-triene	*C. morifolium*	Flowers	1D, 2D NMR; ESIMS	[22]
**9**	chrysanthediol A	*C. morifolium*	Flowers	1D, 2D NMR; EIMS	[17]
**10**	kikkanol D	*C. indicum*	Flowers	1D, 2D NMR; HRESIMS	[23]
**11**	kikkanol D monoacetate	*C. indicum*	Flowers	1D, 2D NMR; HRFABMS	[23]
**12**	kikkanol E	*C. indicum*	Flowers	1D, 2D NMR; HRESIMS; Mosher	[23]
**13**	1*β*-hydroxy-4(15),5*E*,10(14)-germacratriene	*C. indicum*	Flowers	1D, 2D NMR; ESIMS	[11]
**14**	chrysanthemumin I	*C. indicum*	Flowers	1D, 2D NMR; ESIMS	[11]
**15**	chrysandiol	*C. morifolium*			[24]
**16**	chrysanthemumin H	*C. indicum*	Flowers	1D, 2D NMR; HRESIMS; X-ray; ECD	[11]
**17**	1*β*,3*β-*dihydroxygermacra-4Z,10(14)-dien-6*β*,7*α*,11*α*H-12,6-olide	*C lavandulifolium*	Aerial parts	1D, 2D NMR; EIMS	[18]
**18**	1*β*-hydroperoxy-3*β-*hydroxygermacra-4Z,10(14)-dien-6*β*,7*α*,11*α*H-12,6-olide	*C lavandulifolium*	Aerial parts	1D, 2D NMR; EIMS	[18]
**19**	zawadskinolide D	*C. zawadskii*	Aerial parts	1D, 2D NMR; HRFABMS	[19]
**20**	zawadskinolide E	*C. zawadskii*	Aerial parts	1D, 2D NMR; HRCIMS	[19]
**21**	zawadskinolide F	*C. zawadskii*	Aerial parts	1D, 2D NMR; HRCIMS	[19]
**22**	zawadskinolide A	*C. zawadskii*	Aerial parts	1D, 2D NMR; HREIMS	[19]
**23**	zawadskinolide B	*C. zawadskii*	Aerial parts	1D, 2D NMR; HREIMS	[19]
**24**	zawadskinolide C	*C. zawadskii*	Aerial parts	1D, 2D NMR; HREIMS	[19]
**25**	kikkanol F	*C. indicum*	Flowers	1D, 2D NMR; HRESIMS	[23]
**26**	kikkanol F monoacetate	*C. indicum*	Flowers	1D, 2D NMR; HRESIMS	[23]
**Eudesmane**					
**27**	chrysanthemumol I	*C. indicum*	Flowers	1D, 2D NMR; HRESIMS	[21]
**28**	chrysanthemumol J	*C. indicum*	Flowers	1D, 2D NMR; HRESIMS	[21]
**29**	eudesm-4(14)-ene-3*α*,11-diol	*C. indicum*	Flowers	1D, 2D NMR; ESIMS	[21]
**30**	(3*β*,5*α*,6*β*,7*β*,14*β*)-eudesmen-3,5,6,11-tetrol	*C. indicum*	Flowers	1D NMR; X-ray	[25]
**31**	chrysanthemumin A	*C. indicum*	Flowers	1D, 2D NMR; HRESIMS	[11]
**32**	5α-hydroxy-*β*-eudesmol	*C. indicum*	Flowers	1D, 2D NMR; ESIMS	[11]
**33**	chrysanthemumin D	*C. indicum*	Flowers	1D, 2D NMR; HRESIMS	[11]
**34**	chrysanthemumin E	*C. indicum*	Flowers	1D, 2D NMR; HRESIMS	[11]
**35**	chrysanthemumin B	*C. indicum*	Flowers	1D, 2D NMR; HRESIMS	[11]
**36**	7-epi-1*β*-hydroxy-*β*-eudesmol	*C. indicum*	Flowers	1D, 2D NMR; HRESIMS	[26]
**37**	chrysanthemol	*C. indicum*	Flowers	1D, 2D NMR; HREIMS	[27]
**38**	chrysanthemumin F	*C. indicum*	Flowers	1D, 2D NMR; HRESIMS; ECD	[11]
**39**	ligucyperonol	*C. indicum*	Flowers	1D, 2D NMR; HRESIMS	[11]
**40**	chrysanthemumol K	*C. indicum*	Flowers	1D, 2D NMR; HRESIMS	[21]
**41**	canusesnol E	*C. indicum*	Flowers	1D, 2D NMR; ESIMS	[21]
**42**	chrysanthemumin C	*C. indicum*	Flowers	1D, 2D NMR; HRESIMS	[11]
**43**	*β*-dictyopterol	*C. morifolium*	Flowers	1D, 2D NMR; HRESIMS	[17]
**44**	7-epi-eudesm-4(15),11(13)-diene-1*β*,3*β*-diol	*C. indicum*	Flowers	1D, 2D NMR; HRESIMS	[26]
**45**	intermedeol	*C. indicum*	Flowers	1D, 2D NMR; ESIMS	[11]
**46**	cyperusol C	*C. morifolium*	Flowers	1D, 2D NMR; ESIMS	[22]
**47**	chrysantiloboside	*C. zawadskii*	Aerial parts	1D, 2D NMR; FABMS	[19]
**48**	oplodiol 1-*O*-*β*-D-glucopyranoside	*C. zawadskii*	Aerial parts	1D, 2D NMR; HREIMS	[19]
**49**	kikkanol C	*C. indicum*	Flowers	1D, 2D NMR; HREIMS; Mosher	[28]
**50**	kikkanol B	*C. indicum*	Flowers	1D, 2D NMR; HREIMS; Mosher	[28]
**51**	kikkanol A	*C. indicum*	Flowers	1D, 2D NMR; HREIMS; Mosher	[28]
**52**	eudesm-4(15)-ene-1*β*,6*α*-diol	*C. indicum*	Flowers	1D, 2D NMR; ESIMS	[11]
**Guaianolide**					
**53**	angeloylcumambrin B	*C. ornatum* *C. indicum* *C. zawadskii*	Whole herbs Flowers Flowers	1D, 2D NMR; ESIMS 1D, 2D NMR; ESIMS 1D, 2D NMR; ESIMS	[29] [11] [30]
**54**	cumambrin A	*C.* *ornatum* *C. indicum* *C. zawadskii*	Whole herbs Flowers Flowers	1D, 2D NMR; ESIMS 1D, 2D NMR; ESIMS 1D, 2D NMR; ESIMS	[29] [11] [30]
**55**	cumambrin-B	*C.* *ornatum*	Whole herbs	1D, 2D NMR; ESIMS	[29]
**56**	chrysanolide G	*C. indicum*	Aerial parts	1D, 2D NMR; HRESIMS	[31]
**57**	tigloylcumambrin B	*C. indicum*	Aerial parts	1D NMR	[31]
**58**	chrysanolide F	*C. indicum*	Aerial parts	1D, 2D NMR; HRESIMS	[31]
**59**	8-tigloylchrysanolide F	*C. indicum*	Aerial parts	1D, 2D NMR; HRESIMS	[31]
**60**	chrysanolide B	*C. indicum*	Aerial parts	1D NMR	[31]
**61**	10*α*-hydroxy-8*α*-*O*-(*β*-D-glucopyranosyl)-1*α*H,5*α*H, 6*β*H,8*β*H,7*α*H,11*β*H,11*α*-methylguaia-3-enolide	*C. indicum*	Flowers	1D, 2D NMR; HRESIMS	[32]
**62**	chrysanthemumin J	*C. indicum*	Flowers	1D, 2D NMR; HRESIMS; ECD	[11]
**63**	chrysanolide H	*C. indicum*	Aerial parts	1D, 2D NMR; HRESIMS; ECD	[31]
**64**	8-angeloylchrysanolide H	*C. indicum*	Aerial parts	1D, 2D NMR; HRESIMS	[31]
**65**	chrysanthguaianolactone D	*C. morifolium*	Flowers	1D, 2D NMR; HRESIMS	[33]
**66**	chrysanthguaianolactone C	*C. morifolium*	Flowers	1D, 2D NMR; HRESIMS	[33]
**67**	3*α*,4*α*,10*β*-trihydroxy-8*α*-acetoxyguai-1,11(13)-dien-6*α*,12-olide	*C. morifolium*	Flowers	1D NMR	[33]
**68**	3*α*,4*α*,10*β*-trihydroxy-8*α*-acetoxy-11*β*H-guai-1-en-6*α*,12-olide	*C. morifolium*	Flowers	1D NMR	[33]
**69**	1*α*,3*α*,4*β*-trihydroxy-8*α*-acetoxy-9-en-6*α*,12-olide	*C. morifolium*	Flowers	1D NMR	[33]
**70**	chrysanthguaianolactone E	*C. morifolium*	Flowers	1D, 2D NMR; HRESIMS	[33]
**71**	8*α*-(angelyloxy)-3*β*,4*β*-dihydroxy-5*α*H,6*β*H,7*α*H, 11*α*H-guai-1(10)-en-12,6-olide	*C. morifolium*	Flowers	1D NMR	[33]
**72**	indicumolide A	*C. indicum*	Flowers	1D, 2D NMR; HREIMS	[34]
**73**	indicumolide B	*C. indicum*	Flowers	1D, 2D NMR; HREIMS	[34]
**74**	chrysanolide I	*C. indicum*	Aerial parts	1D, 2D NMR; HRESIMS; ECD	[31]
**75**	3*β*,4*α*-dihydroxy-8*α*-angelyloxy-1(10),11(13)-dien-6*β*,12-olide	*C. morifolium*	Flowers	1D, 2D NMR; HRESIMS	[35]
**76**	11,13-dehydrodesacetylmatricarin	*C. indicum*	Flowers	1D, 2D NMR; ESIMS	[11]
**77**	matricarin	*C. indicum*	Flowers	1D, 2D NMR; ESIMS	[11]
**78**	8*β*-angeloyloxy-1*β*,4*β*,10*β*-trihydroxy-guai-2-en-6*α*,12-olide	*C. indicum*	Aerial parts	1D, 2D NMR; EIMS	[36]
**79**	chrysanthguaianolactone B	*C. indicum*	Flowers	1D, 2D NMR; HRESIMS	[37]
**80**	(3*α*,6*α*,8*α*)-8-tigloyl-3,4-epoxyguai-1(10)-eno-12,6-lactone	*C. indicum*	Flowers	1D NMR	[37]
**81**	Angeloylajad	*C. indicum*	Aerial parts	1D NMR; EIMS	[38]
**82**	arteglasin A	*C. indicum*	Aerial parts	1D NMR; EIMS	[38]
**83**	guaianolide ajadin	*C. indicum*	Aerial parts	1D NMR; EIMS	[38]
**84**	apressin	*C. indicum*	Flowers	1D NMR	[37]
**85**	athanadregeolid	*C. indicum*	Flowers	1D NMR	[37]
**86**	chrysanthemulide H	*C. indicum*	Aerial parts	1D, 2D NMR; HRESIMS; ECD	[39]
**87**	8-tigloyldesacetylezomontanin	*C. indicum*	Aerial parts	1D NMR	[39]
**88**	10*α*-hydroxy-1*α*,4*α*-endoperoxy-guaia-2-en-12,6*α*-olide	*C. morifolium*	Flowers	1D, 2D NMR; HRESIMS	[40]
**89**	chrysanthemulide F	*C. indicum*	Aerial parts	1D, 2D NMR; HRESIMS; ECD	[39]
**90**	chrysanthemulide G	*C. indicum*	Aerial parts	1D, 2D NMR; HRESIMS; ECD	[39]
**91**	10-epiajafinin	*C. indicum*	Aerial parts	1D NMR	[39]
**92**	chrysanthemulide A	*C. indicum*	Aerial parts	1D, 2D NMR; HRESIMS; ECD	[39]
**93**	chrysanthemulide B	*C. indicum*	Aerial parts	1D, 2D NMR; HRESIMS; ECD	[39]
**94**	chrysanthemulide C	*C. indicum*	Aerial parts	1D, 2D NMR; HRESIMS; ECD	[39]
**95**	chrysanthemulide D	*C. indicum*	Aerial parts	1D, 2D NMR; HRESIMS; ECD	[39]
**96**	chrysanthemulide E	*C. indicum*	Aerial parts	1D, 2D NMR; HRESIMS; ECD	[39]
**97**	chrysanthguaianolactone A	*C. indicum*	Flowers	1D, 2D NMR; HRESIMS	[37]
**98**	isoseco-tanapartholide	*C. indicum*	Aerial parts	1D, 2D NMR; ECD	[41]
**99**	(-)-9-angeloyloxy-seco-tanapartholide B	*C. indicum*	Aerial parts	1D, 2D NMR; HRESIMS; ECD	[41]
**100**	(-)-seco-tanapartholide B	*C. indicum*	Aerial parts	1D, 2D NMR; ECD	[41]
**101**	(-)-seco-tanapartholide A	*C. indicum*	Aerial parts	1D, 2D NMR; ECD	[41]
**102**	(+)-seco-tanapartholide A	*C. indicum*	Aerial parts	1D, 2D NMR; ECD	[41]
**103**	(+)-seco-tanapartholide B	*C. indicum*	Aerial parts	1D, 2D NMR; ECD	[41]
**104**	handelin	*C. ornatum* *C. indicum*	Whole herbs Aerial parts	1D, 2D NMR; ESIMS 1D NMR	[29] [31]
**105**	chrysanolide D	*C. indicum*	Aerial parts	1D, 2D NMR; HRESIMS; X-ray	[31]
**106**	chrysanolide E	*C. indicum*	Aerial parts	1D, 2D NMR; HRESIMS; ECD	[31]
**107**	8′-tigloylchrysanolide D	*C. indicum*	Aerial parts	1D, 2D NMR; HRESIMS	[31]
**108**	8-angeloyl-8′-hydroxychrysanolide D	*C. indicum*	Aerial parts	1D, 2D NMR; HRESIMS	[31]
**109**	8,8′-ditigloylchrysanolide D	*C. indicum*	Aerial parts	1D, 2D NMR; HRESIMS	[31]
**110**	8-tigloylchrysanolide D	*C. indicum*	Aerial parts	1D, 2D NMR; HRESIMS	[31]
**111**	artanomalide C	*C. indicum*	Aerial parts	1D NMR	[31]
**112**	-	*C. indicum*	Flowers	1D, 2D NMR; HRESIMS; X-ray	[42]
**113**	chrysanthemulide I	*C. indicum*	Aerial parts	1D, 2D NMR; HRESIMS; ECD	[39]
**114**	chrysanthemulide J	*C. indicum*	Aerial parts	1D, 2D NMR; HRESIMS; ECD	[39]
**115**	chrysanolide C	*C. indicum*	Flowers	1D, 2D NMR; HRESIMS	[43]
**116**	chrysanolide A	*C. indicum*	Flowers	1D, 2D NMR; HRESIMS; ECD	[43]
Other types					
**117**	jinsidajuol A	*C. morifolium*	Flowers	1D, 2D NMR; HRESIMS	[22]
**118**	jinsidajuol B	*C. morifolium*	Flowers	1D, 2D NMR; HRESIMS	[22]
**119**	chrysetuno	*C. indicum*	Aerial parts	1D, 2D NMR; EIMS	[36]
**120**	tunefulin	*C. indicum*	Aerial parts	1D, 2D NMR; EIMS	[36]
**121**	11-hydroxy-1-oxo-4*α*,5*α*,7*β*,10*β*-eremophilane	*C. indicum*	Flowers	1D, 2D NMR; ESIMS	[11]
**122**	indicumolide C	*C. indicum*	Flowers	1D, 2D NMR; HRESIMS	[34]
**123**	oplopanone	*C. indicum*	Flowers	1D NMR; EIMS	[28]
**124**	spathulenol	*C. indicum*	Flowers	1D, 2D NMR; ESIMS	[11]
**125**	caryolane 1,9*β*-dio	*C. indicum*	Flowers	1D NMR; EIMS	[28]
**126**	chrysanthemumin G	*C. indicum*	Flowers	1D, 2D NMR; ESIMS	[11]
**127**	11(7→6)abeo-14-norcarbrane-4,7-dione	*C. indicum*	Flowers	1D, 2D NMR; ESIMS	[11]
**128**	6,8-cycloeudesm-4(15)-en-1-ol	*C. indicum*	Flowers	1D, 2D NMR; ESIMS	[11]
**129**	(4*R*,5*R*)-4,5-dihydroxycaryophyll-8(13)-ene	*C. indicum*	Flowers	1D, 2D NMR; ESIMS	[11]
**130**	*β*-caryophyllene	*C. indicum*	Aerial parts	1D NMR	[44]
**131**	grandiflorolide	*C. grandiflora*	Flowers	1D, 2D NMR; HRESIMS	[45]
**132**	clovanediol	*C. indicum*	Flowers	1D NMR; EIMS	[28]
**133**	-	*C. indicum*	Flowers	1D, 2D NMR; HRESIMS	[42]
**134**	-	*C. indicum*	Flowers	1D, 2D NMR; HRESIMS; X-ray	[42]
**135**	chamazulene	*C. indicum*	Aerial parts	1D NMR	[44]

Note: NMR, nuclear magnetic resonance; HRESIMS, high resolution electrospray ionization mass spectroscopy; ESIMS, electron ionization mass spectrometry; X-ray, X-ray crystallography; ECD, electronic circular dichroism.

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
