# Peer review of "Chemistry and Pharmacological Activity of Sesquiterpenoids from the Chrysanthemum Genus"

_molecules, 2021, doi:10.3390/molecules26103038_

Round 1

Reviewer 1 Report

The authors of "Chemistry and pharmacological activity of sesquiterpenoids from the Chrysanthemum genus" have done a nice review of the subject for this important genus.

The main thing lacking in this paper is the information on identification, or rather confirmation and structure resolving methods used by respective authors in identifying, resolving and confirming these (at some points rather complex) structures. I recommend providing such information by simply adding one column before Ref. column in Table 1 and populating it with abbreviations like NMR ((1H, 13C, 1D, 2D, etc.)), HRMS which would suffice and greatly improve this review.

Other minor comments are marked in yellow in pdf attached.

Reviewer 2 Report

First, about the positive aspects of this review. The review is indeed for the first time summarizing information on the sesquiterpenoids of the Chrysanthemum genus plants. The literature analyzed in this review is covered quite fully, including very recent references.

At the same time, in my opinion, the review lacks some information that would allow a better understanding of this topic.

Namely (recommendatory):

1) Indicate which parts of the plant were studied. The conclusion mentions the aerial part, but I think it would be better to mention this at the beginning. What parts of the plant have the most sesquiterpenoids?

2) Briefly describe the methods of isolation of  sesquiterpenoids from plant materials

3) Add information about which of the described compounds are unique for a given species, or are they not?

In addition, a number of comments arose that require correction:

1) line 74 onwards (75, 83, 93, …). What is ANTHRACENE!? Sesquiterpenoids have nothing to do with anthracene.

2) line 91, fig 3, images of substitiuents, SEN = PROP, clarify the structure

3) “The structures” should be replace to “examples of structures”

4) line 66 “complex” is a wrong word

5) the biosynthetic scheme (fig 7) should be presented more precisely and in detail. If there are different hypotheses on this topic - indicate. As far as can be found from the literature (Damian Paul Drew et al. Guaianolides in apiaceae: perspectives on pharmacology and biosynthesis Phytochemistry Reviews volume 8, pages581-599 (2009); Andreas Schall et al. Synthesis of Biologically Active Guaianolides with a trans-Annulated Lactone Moiety https://doi.org/10.1002/ejoc.200700880 EurJOC Volume2008, Issue14 Pages 2353-2364), biosynthesis is supposed via epoxides.

6) line 181. Section 4.3 - it is not clear whether the antibacterial activity was tested for sesquiterpenoids isolated from Chrysanthemum genus or from other plants.

7) line 215, isolates are extracts or individual compounds?

8) line 217 “antidiabetic and antiobesity activities” - there is no information about this type of activity in the review text. Need to add.

The conclusion as a whole reflects the content of the review and is written in sufficient detail.

Round 2

Reviewer 1 Report

The authors have amended their manuscript according to comments.

Reviewer 2 Report

The manuscript was corrected and improved according to comments. All comments are taken into account.